# Consequences of Type-2 diabetes mellitus and Malaria co-morbidity on sperm parameters in men; a case-control study in a district hospital in the Ashanti Region of Ghana

**Ratif Abdulai[1,2], Samuel Addo Akwetey[3], Olayinka Oladunjoye Ogunbode[1,2], Benjamin Aboagye[4]***

**1** Pan African University Institute of Life and Earth Sciences-Including Health and Agriculture, University of Ibadan, Ibadan, Nigeria, **2** Department of Obstetrics and Gynaecology, College of Medicine, University of Ibadan/University College Hospital, Ibadan, Nigeria, **3** Department of Clinical Microbiology, School of Medicine, University for Development Studies, Tamale, Ghana, **4** Department of Forensic Sciences, School of Biological Sciences, College of Agriculture and Natural Sciences, University of Cape Coast, Cape Coast, Ghana

* baboagye@ucc.edu.gh

## Abstract

Type-2 diabetes mellitus (T2DM) and malaria infection are highly prevalent in Africa particularly, in the Sub-Saharan Region. A greater number of people in the Ghanaian population who have T2DM are also reported to harbor malaria parasites. This study aimed to investigate the influence of T2DM & Malaria co-morbidity on sperm parameters among patients in the Ashanti Region of Ghana. This hospital-based cross-sectional analytic case-control study comprised 254 adult male study participants comprising 80 T2DM & Malaria co-morbidity, 80 T2DM only, and 94 normal controls. A blood sample (10mL) was drawn from each participant to measure FBG, HbA1c levels, Testosterone levels, Total cholesterol, and determination of Malaria parasite density. Seminal fluid was also collected from each participant for semen analysis. Sperm kinetics of the T2DM & Malaria co-morbidity group particularly; total motility, rapid progressive motility, and slow progressive motility were negatively affected compared to both T2DM only (p<0.0001) and the Normal control (p<0.0001). Normal sperm morphology was significantly affected in the co-morbidity group compared to T2DM only (p<0.0001). Sperm vitality was also statistically significantly reduced in the T2DM & Malaria co-morbidity than in T2DM only (t $_{(64)}$ = -8.62; p<0.001). There was a significant decline in testosterone levels in the T2DM & Malaria co-morbidity group than in the T2DM only (p<0.0001) and the control (p <0.0001). In conclusion, T2DM and malaria infection have a stronger propensity to alter sperm morphology and lower sperm motility and vitality.

**Data Availability Statement:** All relevant data are within the paper and its Supporting Information files.

**Funding:** RA. Grant funding was part a scholarship package and there was no specific grant ID. The study was funded by the African Union Commission. https://au.int/en/commission The funders had no role in study design, data collection and analysis, decision to publish, or preparation of the manuscript.

**Competing interests:** The authors have declared that no competing interests exist.

**Abbreviations:** T2DM, Type-2 Diabetes Mellitus; FBG, Fasting Blood Glucose; HbA1c, Glycated Haemoglobin; SPSS, Statistical Package for Social Sciences; EDH, Effiduase Government Hospital; SBP, Systolic blood pressure; DBS, Diastolic blood pressure; RDT, Rapid Diagnostic Test.

## Introduction

Type-2 diabetes mellitus (T2DM) and malaria continue to affect millions of people worldwide. Whereas T2DM can be seen to be a global issue, malaria seems to be more prevalent in relatively less developed economies of the globe [1].

T2DM is a complicated chronic metabolic disorder characterized by hyperglycemia, which often results from defects in insulin secretion, insulin resistance, or both [2]. It is known that T2DM affects the fertility of males. Infertility is a disease of the reproductive system which is defined by the inability of couples to conceive after 12 months or more of unprotected regular sexual intercourse [3]. Male infertility could be caused by poor sperm parameters [4] such as reduced sperm motility, sperm concentration and morphology [4–6]. In humans, abnormal sperm quality accounts for about 40–50% of all cases of infertility affecting about 7% of all men [7, 8]. Male fertility problems, related to T2DM are anticipated to increase proportionally as the number of patients living with T2DM continues to rise since the majority of patients are diagnosed when they are of childbearing age [9]. T2DM can cause hormonal disruptions which may negatively impact penile erection and ejaculation, coupled with abnormal sperm production in men [6, 10, 11].

It is also known that T2DM makes it favorable for malaria parasites to survive in the blood of patients living with T2DM. Studies conducted in Ghana revealed that patients with T2DM had a 46% increased risk of *P. falciparum* infection [12] and that malaria parasite infection is associated with an increased rate of gluconeogenesis [13]. Recent evidence from human research also points to the possibility that both children and adults with malaria parasite infections develop insulin resistance, most likely as a result of interactions with inflammatory, oxidative stress, and lipid metabolic indicators [14, 15]. A preclinical study conducted by Ojezele et al. [16] revealed a significant decrease in sperm motility, normal morphology, concentration, vitality, and serum testosterone in the untreated malaria-infected group when compared to the controls [16]. Kim & Moley's investigation in 2008, also revealed lower sperm concentration and reduced levels of serum testosterone in diabetic rats [17].

Because the increased prevalence of T2DM also enhances the survival of malaria parasites in the host blood, and both conditions are known to have an impact on sperm health and sperm production, it is, therefore, necessary to investigate the influence of T2DM and Malaria co-morbidity on sperm parameters. This study investigated the influence of T2DM and Malaria co-morbidity on sperm parameters in the Ashanti Region of Ghana.

## Subjects and methods

### Study design and study setting

This hospital-based cross-sectional analytic case-control study was conducted at Effiduase Government Hospital (EDH), located in the Sekyere East district, in the Ashanti Region of Ghana. The total land area of the Sekyere East district is about 730.05 square kilometres and lies between latitude: 6˚45"- 7˚32N" and longitude: 0˚22W". The majority of the inhabitants of this district are traders and other professional jobs. The Effiduase Government Hospital is a 150 bed capacity district hospital and serves as the main referral facility in the district and its neighbouring villages. This hospital records about 600 cases of diabetes annually, with 548 of them being T2DM (per the outpatient department's report).

### Study participants

The study participants comprised 254 consented adult males, 160 of them had registered and were receiving treatment at the diabetic clinic of the Effiduase Government Hospital and 94

control participants comprised of healthy blood donors, members of staff of EDH, and those visiting their relatives on hospital admission. Out of the 160 Type-2 diabetic men, 80 of them had only T2DM and the other 80 had T2DM co-morbid with malaria infection. The study excluded patients on androgens, steroids, patients with a history of chronic renal failure, prostate cancer, prostatectomy, and castrated men. The exclusion criteria for the health group was based on measurement of baseline fasting blood glucose (FBG) $\geq$7.0 mmol/L, and HbA1c >6.5% and no malaria parasite identification [17]. Data collection started in January 2022 and ended in June 2022. Data entry and analyses were done between July and August 2022. All laboratory analyses were conducted at the MediLab Diagnostic Services, Kumasi-Ashanti.

## Measurement of anthropometric variables

A trained registered nurse measured blood pressure (BP) using a mercury sphygmomanometer and a stethoscope. The mean results were reported in mmHg after duplicate measurements with a 5-minute rest period in between them. The height of study participants to the nearest centimetre without shoes and weight to the nearest 0.1 kg in light clothing was measured with a portable stadiometer and an electronic weighing scale respectively. Body mass index (BMI) was calculated by dividing weight (kg) by height squared ($m^2$). Following anthropometric measurement, a well-structured questionnaire and patient's medical records were used to obtain relevant information on demographic variables such as age, sex, occupation, monthly income, gender, marital status and educational level. Participants' knowledge of malaria was also assessed as well as information on the participant's lifestyles.

## Blood sample collection

Ten (10ml) of blood was drawn via a sterile venipuncture between 7:00 am and 10:00 am after overnight fasting. 1 ml of the blood was dispensed into a fluoride oxalate tube for determination of fasting blood glucose, 4 ml into a tube containing EDTA for determination of Glycated Haemoglobin and malaria parasite density, and 5 ml of each blood sample was transferred into serially numbered plain tubes and centrifuged immediately at 5000 revolutions per minute for five minutes to obtain the serum. The clear serums were then pipetted into clean and dry test tubes and separated into aliquots and frozen at -40 C until analyzed for testosterone. Hormonal estimation was determined by a chemiluminescent enzyme immunoassay system using an automated Beckman Coulter immunoassay analyzer (MediLab Diagnostic Services Ltd, Bantama opposite Komfo Anokye Teaching Hospital, Kumasi-Ghana).

## Fasting blood glucose (FBG) assay

Type-2 Diabetes Mellitus was diagnosed through laboratory assessment based on current WHO diagnostic criteria (FBG $\geq$7.0 mmol/L or 126mg/dl) and HbA1c >6.5% [17] and confirmed through a physician's recommendations [18]. The Accu-Chek Advantage Blood Glucose Monitoring System (AC; Roche Diagnostics, Indianapolis, IN) was used to analyze blood samples for FBG. The glucometer test strip was inserted into the glucometer device to read the test strip code by the device. A micropipette was then used to pick about 20uL of blood from the fluoride oxalate tube and drop it onto the test strip to measure FBG and determination of diabetes mellitus.

## Determination of malaria parasite and density

A blood sample from each patient was spread out as a thick and thin blood smear, stained with 3ml of 10% Romanovsky stain (Giemsa), and examined with a 100X oil immersion objective.

Visual criteria were used to detect malaria parasites. The stained slides were examined and parasite density was determined by parasite count and those with parasite density of 160/uL were included in the study.

## Laboratory assay (Total testosterone, glycated haemoglobin, lipid profile, and seminal fluid analyses)

Refrigerated serum samples were brought to room temperature before analyses were done. Total cholesterol was analyzed using the fully automated Chemistry analyzer (Beckman Coulter AU480). Total testosterone was determined by a chemiluminescent enzyme immunoassay system (CLEIA) using an automated Beckman Coulter immunoassay analyzer (Access 2 fully automated Analyzer). The sensitivity and normal range of total testosterone, according to the laboratory, was 3.0–10.0 ng/mL. Whole blood for the glycated haemoglobin (HbA1c) assay was collected into potassium EDTA test tubes. The blood samples were stored at a temperature between 4 and 8˚C and analyzed within a week. During the analysis, 10 uL of the whole blood was pipetted and mixed with 490 uL of hemolysate reagent in a ratio of 1:50. The solution was then mixed thoroughly until it became homogeneous. It was then analyzed using the Beckman Coulter AU480 fully automated Analyzer. The normal range of HbA1c according to the laboratory is 4.0–5.6%.

## Seminal fluid analysis

Using the WHO laboratory manual [21] for the examination and processing of human semen, 2021, participants were advised to abstain from sexual intercourse for about two [2] to seven [7] days before semen sample collection. Before semen sample collection, participants were advised to observe all hygienic protocols in order not to contaminate the sample. Semen samples were collected directly into a 50-ml polystyrene container (Sterilin, United Kingdom), and analyzed within one hour of expulsion [19]. The semen samples were assessed for semen volume (ml), pH, rapid and slow progressive sperm motility (A and B, %), total motility (A+B, %), non-progressively motile, and non-motile (immotile) sperm. Sperm concentration ($10^6$/mL), total sperm number ($10^6$/ejaculate), sperm vitality (live sperm, %), sperm morphology; normal forms, NF, (%) and abnormal forms (%) were also counted during the microscopic examination.

## Ethical consideration and informed consent

This research was conducted based on the Helsinki Declaration and the study protocol, consent forms, and participant information materials were reviewed and approved by the University of Ibadan/University College Hospital Ethics Committee (UI/EC/21/0720). Also, approval was obtained from the Ghana Health Service, Ashanti Regional Directorate, Sekyere East District Health Directorate, and the Effiduase Government Hospital's Research and Ethics committee. Written informed consent to participate in the study and to publish was sought after the aims and objectives of the study had been thoroughly explained to them. Participants either signed or thumb printed to give their consent before the commencement of the study, and they were assured of the confidentiality of their data.

## Statistical analysis

Statistical analysis was conducted using the Statistical Package for Social Sciences (SPSS) version 22 and graphs were constructed using Graph Pad Prism version 8. The distribution of data was reported as means with standard deviations. Independent samples t-test was

performed to determine the mean difference in sperm vitality of the T2DM-only group and the Type-2 diabetes mellitus & malaria co-morbidity (Cases). One-way ANOVA was conducted to determine the difference in the means of blood and serum variables as well as sperm parameters of T2DM & malaria co-morbidity, T2DM only, and Normal Controls. Tukey's HSD post hoc tests for multiple comparisons of means of measured indices for T2DM & Malaria Co-morbidity, T2DM only, and Normal Controls were also performed to determine any statistically significant difference in the means of measured variables in any of the groups. All statistical analyses were done at a 95% confidence interval, Mean differences were significant at $p < 0.05$.

## Results

### Socio-demographic characteristics of study participants

The study involved only male participants with a mean age was 43.3±12.2years. A greater number of the participants (33.9%) were older than 50 years while those aged 20–29 were the least (18.1%). Most participants were married (56.7%) and 25.2% were single. The majority (63.8%) of the participants had a tertiary level of education. The participant's employment status varied with a greater proportion (54.3%) being professional workers (Doctors, Nurses, Teachers, Bankers, and security persons). Only a few (3.9%) were unemployed. Most of the participants (37%) receive a monthly income between Ghc 1,500 and Ghc 3,000 while 4% of them receive below Ghc 500 (Table 1).

### Lifestyle information of the study participants

Study participants had different lifestyles. Out of the 254 participants, only 3.9% were smokers, about 80.3% did not consume alcohol, and 13.4% were exposed to harmful chemicals by virtue of their work. About half (58.3%) of them exercised regularly with the majority of them (44.9%) exercising 1–3 times a week (Table 2).

### Clinical history of the study participants

About 9.4% of participants had been treated of some sexually transmitted infections within a period of six months prior to the study while 5.5% underwent surgery on their reproductive organs. The majority (98.4%) of them reported they could feel their testicles, and less than 1% were taking medication for reproduction. None of the participants had cryptorchidism, but 6.3% reported they ever had orchitis. About one-third (35.4%) of the participants consult diabetic experts twice a month, and 31.5% have lived with diabetes between 5 and 10 years (Table 3).

### General knowledge and participants' clinical information on malaria

Table 4 shows the study participants' general knowledge and clinical information on malaria. The percentage T2DM patients who were at greater risk of malaria infection was 41.7%. About 89% participants lived in mosquito-infested areas, 44.9% of them were using treated mosquito nets and 65.3% had been diagnosed with malaria infection at least twice with six months prior to the study. Preferred treatment options were Artemether Lumefantrine (59.1%), Artesunate injection (9.4%) and Artemether injection (16.5%). Most of the participants (64.6%) were aware of their sickle cell and genotype status with 30.7% of them being AA while only a few (4.0%) of them had the genotype SC.

**Table 1. Socio-demographic characteristics of study participants.**

| Variable | Diabetes and Malaria | Diabetes | No condition | Total |
|---|---|---|---|---|
| | N (%) | N (%) | N (%) | N (%) |
| **Age group in years** | | | | |
| 20–29 | 4 (5.0) | 4 (5.0) | 38 (40.4) | **46 (18.1)** |
| 30–39 | 14 (17.5) | 12 (15.0) | 40 (42.6) | **66 (26.0)** |
| 40–49 | 20 (25.0) | 26 (32.5) | 10 (10.6) | **56 (22.0)** |
| ≥ 50 | 42 (52.5) | 38 (47.5) | 6 (6.4) | **86 (33.9)** |
| **Marital status** | | | | |
| Single | 10 (12.5) | 4 (5.0) | 50 (53.2) | **64 (25.2)** |
| Married | 58 (72.5) | 46 (57.5) | 40 (42.6) | **144 (56.7)** |
| Separated | 0 (0.0) | 8 (10) | 2 (2.1) | **10 (3.9)** |
| Divorced | 2 (2.5) | 10 (12.5) | 0 (0.0) | **12 (4.7)** |
| Widowed | 10 (12.5) | 10 (12.5) | 0 (0.0) | **20 (7.9)** |
| No Response | 0 (0.0) | 2 (2.5) | 2 (2.1) | **4 (1.6)** |
| **Highest educational level** | | | | |
| Illiterate | 2 (2.5) | 2 (2.5) | 0 (0.0) | **4 (1.6)** |
| Primary | 10 (12.5) | 6 (7.5) | 0 (0.0) | **16 (6.2)** |
| Secondary | 28 (35.0) | 26 (32.5) | 14 (14.9) | **68 (26.8)** |
| Tertiary | 40 (50.0) | 44 (55.0) | 78 (83.0) | **162 (63.8)** |
| No Response | 0 (0.0) | 2 (2.5) | 2 (2.1) | **4 (1.6)** |
| **Monthly income (Ghc)** | | | | |
| <500 | 6 (7.5) | 0 (0.0) | 4 (4.3) | **10 (4.0)** |
| 500–1500 | 26 (32.5) | 24 (30.0) | 28 (29.8) | **78 (30.7)** |
| 1501–3000 | 34 (42.5) | 36 (45.0) | 24 (25.5) | **94 (37.0)** |
| >3000 | 10 (12.5) | 10 (12.5) | 24 (25.5) | **44 (17.3)** |
| No Response | 4 (5.0) | 10 (12.5) | 14 (14.9) | **28 (11.0)** |
| **Occupation** | | | | |
| Professional | 32 (40.0) | 42 (52.5) | 64 (68.1) | **138 (54.3)** |
| Skilled labour | 16 (20.0) | 12 (15.) | 4 (4.2) | **32 (12.6)** |
| Unskilled manual labour | 14 (17.5) | 18 (22.5) | 6 (6.4) | **38 (15.0)** |
| Unemployed | 6 (7.5) | 0 (0.0) | 4 (4.2) | **10 (3.9)** |
| Others (Pensioner) | 8 (10.0) | 8 (10.0) | 4 (4.2) | **20 (7.9)** |
| Students | 4 (5.0) | 0 (0.0) | 12 (12.8) | **16 (6.3)** |
| **Total** | 80 (100) | 80 (100) | 94 (100) | **254(100)** |

## Blood and serum variables of study participants

Data for blood and serum variables are represented in Fig 1. Fasting blood glucose concentration and glycated haemoglobin levels of the Control group were significantly lower than that of T2DM & Malaria co-morbidity ($p < 0.0001$) and T2DM only ($p < 0.0001$). Testosterone level was higher in the Control group than in the T2DM & Malaria co-morbidity ($p < 0.0001$) and T2DM only ($p < 0.0001$). Compared to the T2DM-only group, the mean testosterone level of the T2DM & Malaria co-morbidity was significantly low ($p < 0.0001$).

## Sperm kinetics of the study groups

Information on sperm kinetics of the study groups is presented in Fig 2. Rapid and Slow progressive motilities of the control group were significantly higher than the T2DM & Malaria co-morbidity ($p < 0.0001$) and the T2DM only ($p < 0.0001$). Total sperm motility, Rapid and Slow

**Table 2. Lifestyle information of the study participants.**

| Variable | Frequency (N = 254) | Percentage (%) |
|---|---|---|
| **Do you smoke (cigarettes, or marijuana)?** | | |
| Yes | 10 | 3.9 |
| No | 244 | 96.1 |
| **Do you drink alcohol?** | | |
| Yes | 50 | 19.7 |
| No | 204 | 80.3 |
| **Do your profession or daily activities expose you to toxic chemicals** | | |
| Yes | 34 | 13.4 |
| No | 220 | 86.6 |
| **Do you exercise regularly?** | | |
| Yes | 148 | 58.3 |
| No | 104 | 40.9 |
| No Response | 2 | 0.8 |
| **How many times do you exercise in a week?** | | |
| 0 | 86 | 33.9 |
| 1–3 | 114 | 44.9 |
| 4–7 | 40 | 15.7 |
| $\geq 8$ | 12 | 4.7 |
| No Response | 2 | 0.8 |

progressive sperm motilities of the T2DM & Malaria co-morbidity group were also significantly lower than the T2DM-only group (p<0.0001). Comparisons of the groups again revealed that the mean non-progressive sperm motility of the Control was significantly higher than the T2DM & Malaria co-morbidity (p = 0.0006), and T2DM only (p = 0.0008). However, there was no statistically significant difference between T2DM & Malaria co-morbidity and T2DM only (P = 0.830). Again, the mean immotile sperm of the Control was significantly lower than the T2DM & Malaria co-morbidity (p<0.0001) and T2DM only (p<0.0001). There was also a significant difference between the T2DM & Malaria co-morbidity group and T2DM only (p<0.0001).

### Sperm count and semen volume of the study groups

Fig 3 shows the means of semen volume, sperm concentration, and total sperm count of the study groups. The mean volume of semen of the Control group was significantly higher than the T2DM & Malaria co-morbidity (p = 0.001) and T2DM only (p = 0.002), however, no statistically significant difference was found between T2DM & Malaria co-morbidity and T2DM only (p = 0.963). Specific comparisons again indicated that the means of sperm concentration and total sperm count of the control group were significantly higher than the T2DM & Malaria co-morbidity (p<0.0001) and T2DM only (p<0.0001). However, no statistically significant difference in sperm concentration and total sperm count was observed between T2DM & Malaria co-morbidity and T2DM only (P = 0.157; p = 0.395) respectively.

**Sperm morphology (Normal and abnormal forms) and sperm vitality.** Results for sperm morphology (Normal and Abnormal forms) and vitality are shown in Fig 4. The mean normal sperm morphology of the Control was significantly higher than the T2DM & Malaria co-morbidity (p<0.0001) and T2DM only (p<0.0001). A significant difference also existed between T2DM & Malaria co-morbidity and T2DM only (p<0.0001). The mean abnormal sperm morphology was higher in the T2DM & Malaria group than in the control (p<0.0001)

**Table 3. Clinical history of the study participants.**

| Variable | Frequency (N = 254) | Percentage (%) |
|---|---|---|
| **Have you received STI treatment in the past 6 months?** | | |
| Yes | 24 | 9.4 |
| No | 230 | 90.6 |
| **Have you had any previous surgery on any reproductive organ?** | | |
| Yes | 14 | 5.5 |
| No | 240 | 94.5 |
| **Can you feel both testicles?** | | |
| Yes | 250 | 98.4 |
| No | 4 | 1.6 |
| **Do you feel any pain in any of your testicles** | | |
| Yes | 2 | 0.8 |
| No | 252 | 99.2 |
| **Taking any medication for reproduction** | | |
| Yes | 2 | 0.8 |
| No | 252 | 99.2 |
| **Have you ever had orchitis (Inflammation of testicles)** | | |
| Yes | 16 | 6.3 |
| No | 238 | 93.7 |
| **Have you ever had cryptorchidism (Undescended testis into the scrotum)** | | |
| Yes | 0 | 0.0 |
| No | 254 | 100.0 |
| **Frequency of consulting diabetic expert** | | |
| None | 78 | 30.7 |
| Once | 78 | 30.7 |
| Twice | 90 | 35.4 |
| More | 8 | 3.2 |
| **Duration of diabetes (Years)** | | |
| <5 | 58 | 22.8 |
| 5–10 | 80 | 31.5 |
| 11–15 | 18 | 7.1 |
| No diabetes | 94 | 37.0 |
| No Response | 4 | 1.6 |

and T2DM only (p<0.0001). Independent-Sample t-test also revealed a statistically significant difference in sperm vitality between T2DM & Malaria co-morbidity and T2DM only. ($t_{(64)}$ = -8.62; p<0.001).

## Discussion

Type-2 diabetes mellitus (T2DM) and malaria infection are highly prevalent in the Sub-Saharan Region. A larger percentage of T2DM patients in the Ghanaian population is also reported to harbour malaria parasites, and these conditions are thought to have a serious impact on sperm health and production. This study sort to investigate the impact of T2DM & Malaria co-morbidity on sperm parameters and male reproductive endocrinology.

The study first looked at the educational status, lifestyle factors, and anthropometric measurements of the study participants. The majority of participants (63.8%) were educated, and had ample knowledge of T2DM and malaria. For this reason, most of them (69.3%) sort

**Table 4. General knowledge and participants' clinical information on malaria.**

| Variable | Frequency (N = 254) | Percentage (%) |
|---|---|---|
| **Diabetes Mellitus people are at risk of malaria infection** | | |
| Yes | 106 | 41.7 |
| No | 148 | 58.3 |
| **Are you living in a mosquito-infested area?** | | |
| Yes | 226 | 89.0 |
| No | 28 | 11.0 |
| **Number of times diagnosed with malaria in the past 6 months** | | |
| None | 38 | 15.0 |
| 1–2 | 166 | 65.3 |
| 3–4 | 50 | 19.7 |
| **What method of test diagnosis was conducted?** | | |
| RDT | 84 | 33.0 |
| Microscopy | 132 | 52.0 |
| No Response | 38 | 15.0 |
| **Do you sleep in a treated mosquito net?** | | |
| Yes | 114 | 44.9 |
| No | 140 | 55.1 |
| **The malaria medication you took for the past 6 months** | | |
| Artemeter Lumefantrine | 150 | 59.1 |
| Artesunate Injection | 24 | 9.4 |
| Artemeter Injection | 42 | 16.5 |
| No Response | 38 | 15.0 |
| **Do you know your sickle cell status?** | | |
| Yes | 194 | 76.4 |
| No | 60 | 23.6 |
| **Do you know your genotype status** | | |
| Yes | 164 | 64.6 |
| No | 90 | 35.4 |
| **What is your genotype** | | |
| AA | 78 | 30.7 |
| AS | 32 | 12.6 |
| AC | 44 | 17.3 |
| SC | 10 | 4.0 |
| No Response | 90 | 35.4 |

healthcare at least once a month, and refrained from alcohol consumption (80.3%). This indicates that they were well-informed about their condition and needed to avoid behavior that could jeopardize their health.

Secondly, T2DM & Malaria co-morbidity resulted in significant decline in testosterone levels compared with other groups. This implies that Malaria has contributed significantly to the reduced testosterone levels in the co-morbidity group. This observation is in agreement with two preclinical studies conducted by Ojezele et al. [16] and Barthelemy et al. [20]. Findings of a case-control study conducted by Esfandiari et al. on the level of circulating steroid hormones in malaria and cutaneous leishmaniasis infections also revealed low testosterone levels in the malaria-infected group when compared to the control population [21].

The low testosterone levels observed in T2DM & Malaria co-morbidity group may be due to the effect of low glycemia on the testicular micro-vessels [22]. Insulin insensitivity may have

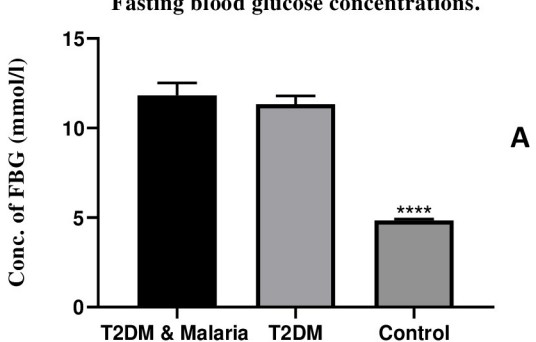

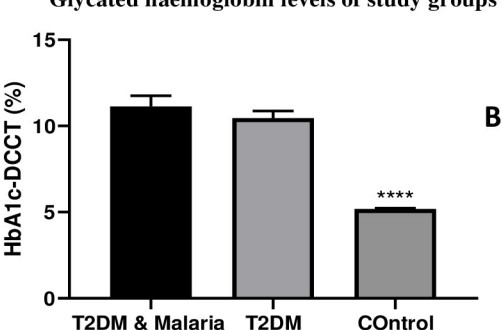

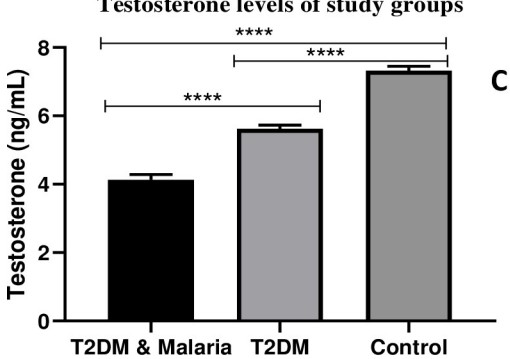

**Fig 1. Blood and serum variables.** Bar charts showing the relationships between Type-2 Diabetes Mellitus (T2DM) & Malaria co-morbidity, Type-2 Diabetes Mellitus (T2DM) only, and the Normal (Control) population based on blood and serum variables. A: Fasting blood glucose (FBG); B: HbA1c-DCCT (%); C: Testosterone levels. Tukey's HSD post hoc test was undertaken to identify mean differences between groups. Data represented Mean±SD, Mean difference was significant at α<0.05; ****p < 0.0001.

negatively affected energy production which is needed for metabolic activities involved in the testosterone secretion process. Low glucose uptake by the hypothalamic and pituitary endocrinolocytes might have also affected gonadotropin secretion hence insufficient testicular stimulation resulting in reduced testosterone secretion [23]. Decreased levels of serum testosterone could also be due to malaria parasite inhibition of testosterone synthesizing enzymes or enhanced induction of testosterone liver conjugating enzymes by the malaria parasites which may have increased the conjugation and subsequent excretion of circulating testosterone, causing serum testosterone levels to decline [16].

Thirdly, results from this study showed significant differences in sperm kinetics. Total sperm motility (A+B) of the T2DM & Malaria co-morbidity and T2DM-only groups were significantly lower when compared to the control group. A significant difference was also recorded between T2DM & Malaria co-morbidity and T2DM only. This indicates that malaria played a very significant role in the reduction of total sperm motility in the T2DM & Malaria co-morbidity as compared to T2DM only. Our findings agree with the results of Condorelli et al. (2018), which stated a significantly reduced total sperm motility, and progressive motility in T2DM adult males compared to the control population [24]. The reduction in sperm motility that was observed in the T2DM & Malaria co-morbidity group might be because of the significant reduction in the levels of testosterone in this group. There is a direct positive

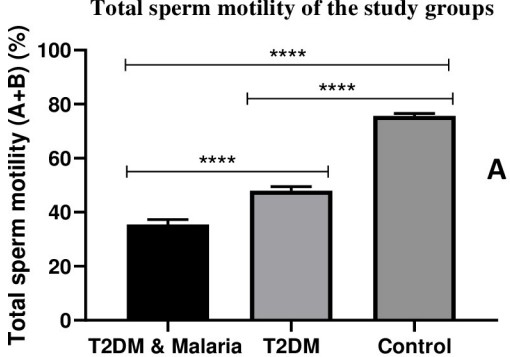

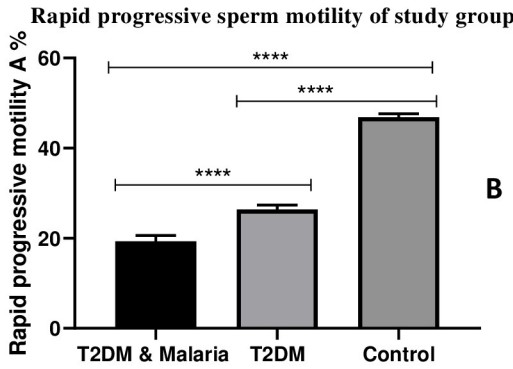

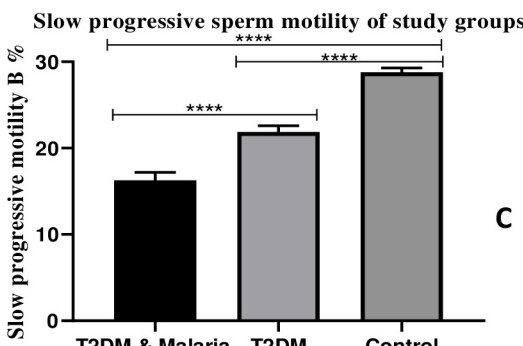

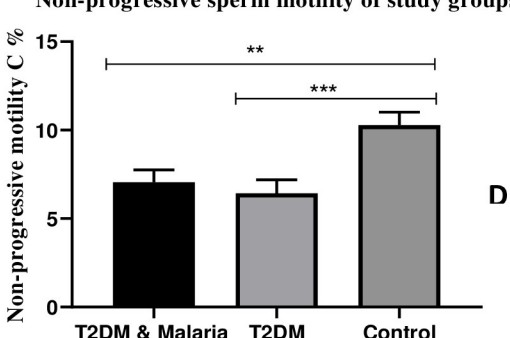

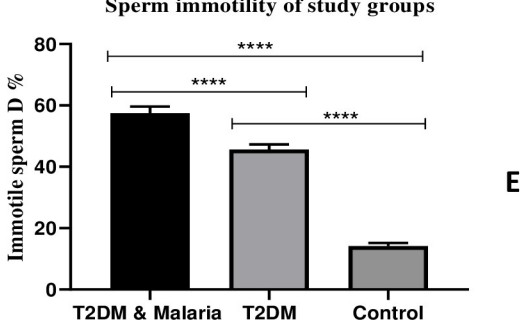

**Fig 2. Sperm kinetics of the study groups.** Bar charts showing the relationships between Type-2 Diabetes Mellitus (T2DM) & Malaria co-morbidity, Type-2 Diabetes Mellitus (T2DM) only, and the Normal (Control) population based on Sperm kinetics. A: Total Motility (A+B); B: Rapid Progressive Motility (A); C: Slow Progressive Motility (B); D: Non-progressive motility (C); E: Immotile sperm (D). Tukey's HSD post hoc test was undertaken to identify mean differences between groups. Data represented Mean±SD, Mean difference was significant at $\alpha < 0.05$; ****$p < 0.0001$; ***$p < 0.001$; **$p < 0.01$.

association between sperm motility and testosterone secretion [25], which partly explains the significant reduction observed in sperm kinetics of the study participants who had both T2DM and malaria infection. Because the function of seminal vesicles that produce a nutrient-rich medium for sperm survival is also testosterone-dependent, low testosterone levels recorded in the T2DM & Malaria co-morbidity group might also be a reason for the reduced

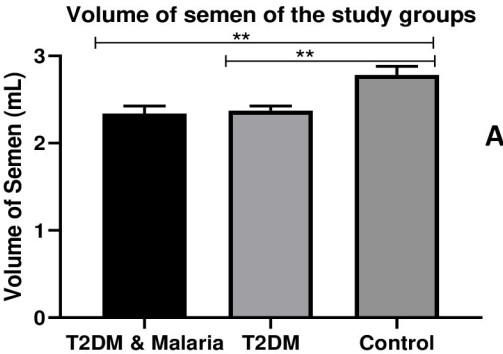

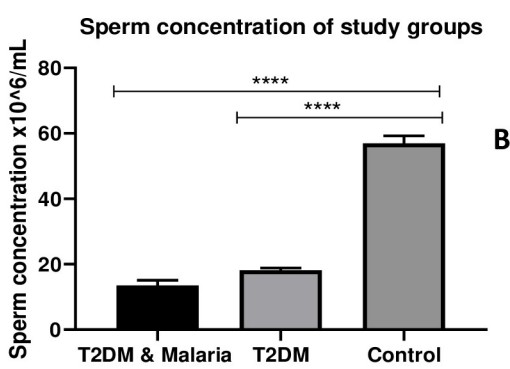

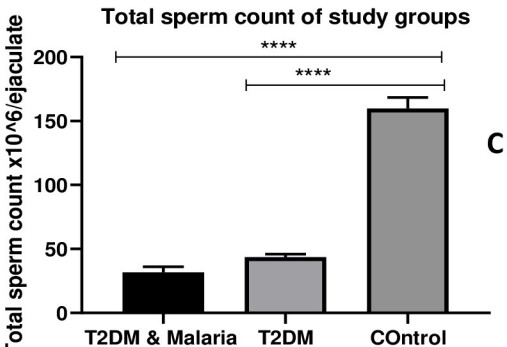

**Fig 3. Sperm count and semen volume of the study groups.** Bar charts showing the relationships between Type-2 Diabetes Mellitus (T2DM) & Malaria co-morbidity, Type-2 Diabetes Mellitus (T2DM) only, and the Normal (Control) population based on sperm volume and sperm count. A: Volume of Semen; B: Sperm Concentration; C: Total Sperm Count. Tukey's HSD post hoc test was conducted to identify mean differences between groups. Data represented Mean±SD, Mean difference was significant at $\alpha < 0.05$; ****$p < 0.0001$; **$p < 0.01$.

sperm motility [26]. There might have been histologic damage caused to the epididymis thus, affecting sperm maturation and transit [5]. In addition, malaria infection may have contributed to the generation of free radicals (toxins) during the parasite's life cycle, especially at the erythrocytic stage, which is very harmful to all cells [27]. The free radicals probably induced changes in the sperm that might have led to reduced sperm motility [27]. Increased oxidative stress associated with T2DM and malaria infection may have induced damage to sperm nuclear and mitochondrial DNA as well as sperm structure, thus affecting sperm motility [16, 28]. This finding is consistent with previous studies that reported that *P. berghei*-induced malaria negatively affected sperm motility [29]. Given that evidence suggests that pyrexia negatively affects sperm motility at some time during spermatogenesis [30], Malaria may have induced fever in the T2DM & Malaria co-morbid group, which probably caused the changes that led to the reduced sperm motility. This further supports the idea that malaria infection co-morbid with T2DM has serious implications for sperm kinetics.

The findings from this study revealed that the mean sperm morphology (Normal forms) of the T2DM & Malaria co-morbidity group was significantly lower when compared to the T2DM-only group and the controls. The decrease observed in the normal forms of sperm morphology of the T2DM & Malaria co-morbidity group agrees with earlier studies that reported a significant reduction in normal sperm morphology [3, 4]. Another study conducted by Condorelli and colleagues reported a decrease in normal forms of sperm. Histologic

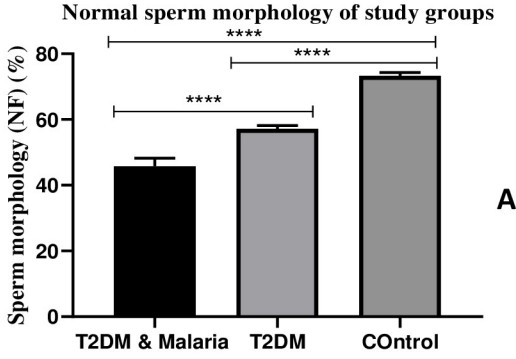

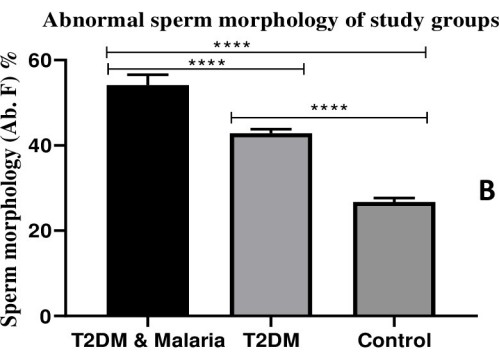

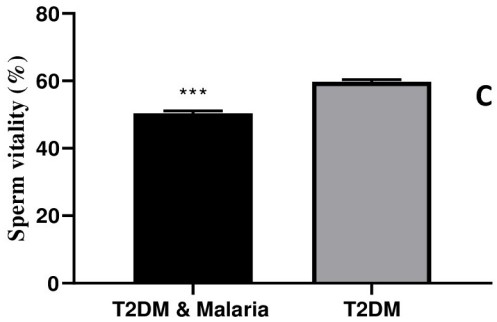

**Fig 4. Sperm morphology (Normal and abnormal forms) and sperm vitality.** Bar charts showing the relationships between Type-2 Diabetes Mellitus (T2DM) and Malaria co-morbidity, Type-2 Diabetes Mellitus (T2DM) only, and the Normal (Control) population based on sperm morphology and vitality. A: Sperm Morphology (Normal forms); B: Sperm Morphology (Abnormal forms); C: Sperm Vitality. Tukey's HSD post hoc test was carried out to identify mean differences between groups. Data represented Mean±SD, Mean difference was significant at α<0.05; ****p < 0.0001; ***p < 0.001.

damage caused by oxidative stress and generation of free radicals may have caused significant damage to sperm thus, affecting sperm morphology.

Lastly, findings of our study also showed a statistically significant difference in sperm vitality between the T2DM & Malaria co-morbidity group and the T2DM only. This implies that malaria contributed significantly to the observed reduced sperm vitality in the T2DM & Malaria co-morbidity group. Our findings are consistent with earlier studies that reported that *P. berghei*-induced malaria infection and T2DM caused an inflammatory reaction with increased oxidative stress, resulting in decreased sperm vitality and increased sperm DNA fragmentation. One of the reasons for reduced sperm vitality in the T2DM & Malaria co-morbidity group could have been the low testosterone levels observed in this group. Testosterone plays a very important role in sperm production and differentiation, it is also needed to maintain physiological processes involved in sperm maturation so a reduction in the levels of testosterone is more likely to affect sperm vitality negatively. More to this, seminal vesicles that produce nutrient-rich medium to ensure the survival of sperm are also testosterone dependent, and therefore, reduced testosterone levels could as well reduce sperm vitality. The morphology of sperm is very important for its survival. It could also be that malaria parasites may have increased the accumulation of free radicals which probably affected their morphology negatively hence their survival.

Recall bias, particularly concerning the lifestyles of the participants could be a limitation of our study. However, the inferential statistics was conducted on most measured parameters. Majority of the study participants were $\geq 50$ years old (33.9%). To help address this limitation, we recommend that future research take into account study only participants under 40 years of age. In the face of these limitations, this study gives significant data on the consequences of T2DM and malaria co-morbidity on sperm parameters among Ghanaian men with T2MD co-morbid with malaria.

## Conclusion

The results of this study showed that co-morbidity of T2DM and malaria has a greater tendency to damage sperm morphology and reduce sperm motility and vitality. Participants in the study who had both T2DM and malaria infection experienced negative effects on all these parameters. Additionally, Serum testosterone was also significantly reduced in this group. This further suggests that co-morbidity of T2DM and malaria has grave consequences and therefore has a high tendency to contribute to male infertility. It is necessary to conduct more research to bring out a deeper understanding of the interaction and biological mechanisms by which plasmodium parasites interact with T2DM to impair sperm quality. Because T2DM makes it favourable for malaria parasites to survive in the body/blood of affected individuals, it is also recommended for the Ghanaian government and the African Union invest in malaria vaccine research to help vaccinate people against malaria to mitigate its impact on public health.

## Supporting information

**S1 Checklist. STROBE statement—checklist of items that should be included in reports of *case-control studies.***
(DOC)

**S1 Table. Means and standard deviations of blood and serum variables among study groups (T2DM & Malaria co-morbidity, T2DM only and the control).** T2DM & Malaria Co-morbidity Group = participants who had both Type-2 diabetes mellitus and malaria infection, T2DM only = participants who had only Type-2 diabetes mellitus and No T2DM & No Malaria Group = the control population, thus, participants who had neither Type-2 diabetes mellitus nor malaria infection, FBG = Fasting Blood Glucose, HbA1c-DCCT = Glycated haemoglobin.
(DOCX)

**S2 Table. Means and standard deviations of sperm parameters among study groups (T2DM & Malaria co-morbidity, T2DM only and the control.** T2DM & Malaria Co-morbidity Group = participants who had both Type-2 diabetes mellitus and malaria infection, T2DM only = participants who had only Type-2 diabetes mellitus and No T2DM & No Malaria Group = the control population, thus, participants who had neither Type-2 diabetes mellitus nor malaria infection.
(DOCX)

**S3 Table. Comparison of mean variables using ANOVA between the study groups (T2DM & Malaria co-morbidity, T2DM only, and the control population).** T2DM & Malaria Co-morbidity Group = participants who had both Type-2 diabetes mellitus and malaria infection, T2DM only = participants who had only Type-2 diabetes mellitus and No T2DM & No Malaria Group = the control population. Data represented Mean±SD. Mean difference was significant at α<0.05. * Statistically significant difference between T2DM & Malaria Co-

morbidity Group, T2DM Only and the Control population.
(DOCX)

**S4 Table. Post hoc test; Tukey's HSD multiple comparisons of means of measured variables between T2DM & Malaria co-morbidity and T2DM only.** T2DM & Malaria Co-morbidity Group = participants who had both Type-2 diabetes mellitus and malaria infection, T2DM only = participants who had only Type-2 diabetes mellitus; Data represented Mean ±SD. Mean difference was significant at $\alpha < 0.05$. * Statistically significant difference between T2DM & Malaria Co-morbidity Group and T2DM Only.
(DOCX)

**S5 Table. Post hoc test; Tukey's HSD multiple comparisons of means of measured variables between T2DM & Malaria co-morbidity and the control.** T2DM & Malaria Co-morbidity Group = participants who had both Type-2 diabetes mellitus and malaria infection and No T2DM & No Malaria Group = the control population. Data represented Mean±SD. Mean difference was significant at $\alpha < 0.05$. * Statistically significant difference between T2DM & Malaria Co-morbidity Group and Control Group.
(DOCX)

**S6 Table. Post hoc test; Tukey's HSD multiple comparisons of means of measured variables between T2DM only and the control.** T2DM only = participants who had only Type-2 diabetes mellitus and No T2DM & No Malaria Group = the control population. Data represented Mean±SD. Mean difference was significant at $\alpha < 0.05$. * Statistically significant difference between T2DM Only and the Control Group.
(DOCX)

## Acknowledgments

We express our sincere appreciation for the support and contributions of the staff of Effiduase Government Hospital, MediLab Diagnostic Services, Pan African University Institute of Life and Earth Sciences-Including Health and Agriculture (PAULESI) and all the study participants.

## Author Contributions

**Conceptualization:** Ratif Abdulai.

**Data curation:** Ratif Abdulai, Samuel Addo Akwetey, Olayinka Oladunjoye Ogunbode.

**Formal analysis:** Ratif Abdulai, Samuel Addo Akwetey.

**Investigation:** Ratif Abdulai.

**Methodology:** Ratif Abdulai.

**Project administration:** Ratif Abdulai.

**Supervision:** Olayinka Oladunjoye Ogunbode, Benjamin Aboagye.

**Validation:** Samuel Addo Akwetey, Olayinka Oladunjoye Ogunbode.

**Visualization:** Ratif Abdulai, Samuel Addo Akwetey.

**Writing – original draft:** Ratif Abdulai.

**Writing – review & editing:** Ratif Abdulai, Samuel Addo Akwetey, Olayinka Oladunjoye Ogunbode, Benjamin Aboagye.

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
