## [Decision Letter · Decision Letter 0]

18 Apr 2023

PONE-D-23-06530Consequences of type-2 diabetes mellitus and malaria co-morbidity on sperm parameters in men; a case-control study in a district hospital in the Ashanti Region of Ghana.PLOS ONE

Dear Dr. Abdulai,

Thank you for submitting your manuscript to PLOS ONE. After careful consideration, we feel that it has merit but does not fully meet PLOS ONE’s publication criteria as it currently stands. Therefore, we invite you to submit a revised version of the manuscript that addresses the points raised during the review process. Please ensure to respond to the reviewers comments.

We look forward to receiving your revised manuscript.

Kind regards,

Raquel Inocencio da Luz, Phd

Academic Editor

PLOS ONE

Journal Requirements:

4. Please amend the manuscript submission data (via Edit Submission) to include author Dr. Benjamin Aboagye.

6. Please clarify the Tables' numbers uploaded in your PDF file.

Reviewers' comments:

Reviewer's Responses to Questions

**Comments to the Author**

1. Is the manuscript technically sound, and do the data support the conclusions?

Reviewer #1: Yes

Reviewer #2: Partly

2. Has the statistical analysis been performed appropriately and rigorously? 

Reviewer #1: Yes

Reviewer #2: I Don't Know

3. Have the authors made all data underlying the findings in their manuscript fully available?

Reviewer #1: Yes

Reviewer #2: Yes

4. Is the manuscript presented in an intelligible fashion and written in standard English?

Reviewer #1: Yes

Reviewer #2: Yes

5. Review Comments to the Author

Reviewer #1: The paper is quite good but there are some typographical and grammatical errors, the commonest of which is putting periods after headings. These periods (full stops) should be removed. There should be a description of the hospital (is it in an urban or rural area? How many bedded hospital is it? Does it have a diabetes clinic? The type of testosterone assay was not well established under the methodology as the authors did not state if it was free or total or bioavailable testosterone that was measured. Was there any other hormone assay (prolactin, estradiol performed?). Also the references need to be well standardised and seem not to follow any certain citing style/pattern especially those from online (internet) journals.

Reviewer #2: Study reports a very important aspect globally; and mostly in Africa where infertility carries important medical and mostly psychosocial implications. The analysis of outcomes has used standard methods, for both malaria and diabetes; results are very relevant and matched with the research questions.

The design is appropriate, and the different essays have been conducted following the standardized methods. However, the application of the design presents some weaknesses which introduced doubt in the findings concerning introduction of possible bias.

We suggest reworking on following observations and revised all criteria of a good case-control study to increase even more the quality of this study.

The design of the study is quite good, and the conduct of the study was smoothly conducted.

The technical aspects related to equipment’s lab techniques used by the authors team was appreciable; however,

The case-control aspect has been a bit neglected. Several aspects reported in case-control studies were not mentioned here (see methodology and even an aspect of the results)

Using the checklist of reporting good case-control studies, we can realize for example:

1. Assignment of patients in groups of the study?

2. Analysis not detailed, stratification-logistic regression, which assumptions?

3. Socio-demographic data is not presented stratified by groups (case/controls) to get an idea about similarities/differences between the two groups.

4. Not considerations of most of the key points used in critical appraisal checklist of a case-control studies

see: Joanna Briggs Institute 2017 Critical Appraisal Checklist for Case Control Studies (jbi.global): https://jbi.global/sites/default/files/2019-05/JBI_Critical_Appraisal-Checklist_for_Case_Control_Studies2017_0.pdf

6. PLOS authors have the option to publish the peer review history of their article (what does this mean?). If published, this will include your full peer review and any attached files.

Reviewer #1: **Yes: **Prof Olufemi Fasanmade

Reviewer #2: **Yes: **no

---

## [Author Response · Author response to Decision Letter 0]

1 May 2023

Point-by-Point Responses to Reviewers’ Comments

Date: 01/05/2023

Submission ID: PONE-D-23-06530

Title of Article: Consequences of type-2 diabetes mellitus and malaria co-morbidity on sperm parameters in men; a case-control study in a district hospital in the Ashanti Region of Ghana.

Name of the Corresponding Author: Benjamin Aboagye

Email Address of the Corresponding Author: baboagye@ucc.edu.gh

Dear Editors and Reviewers,

We greatly appreciate the thorough and thoughtful comments provided on our submitted manuscript. We have addressed and revised accordingly all the comments of the reviewer(s). Attached are our detailed responses to their comments. All changes/clarifications in the manuscript have been highlighted in yellow.

We appreciate your consideration of this manuscript. Please find enclosed our revised manuscript. 

Sincerely,

The Authors of the Manuscript: PONE-D-23-06530

POINT-BY-POINT RESPONSES TO REVIEWERS

Reviewer #1

The paper is quite good but there are some typographical and grammatical errors, the commonest of which is putting periods after headings.

Authors Response: We have tried to provide clarification by addressing the issues raised below:

There should be a description of the hospital (is it in an urban or rural area? How many bedded hospital is it? Does it have a diabetes clinic?

Authors Response: The description of the Effiduase Government Hospital has been made in the manuscript. The hospital is a district hospital with 150 bed capacity and also has a diabetic clinic. (Page 3, lines 72-79)

The type of testosterone assay was not well established under the methodology as the authors did not state if it was free or total or bioavailable testosterone that was measured. Was there any other hormone assay (prolactin, estradiol performed?).

Authors Response: Total testosterone was the type of testosterone measured. No other hormonal assay was done because the study only sort to investigate the influence of T2DM and Malaria co-morbidity on testosterone secretion since it plays a direct role in spermatogenesis, sperm maturation and overall sperm health. (Page 7, line 130, lines 134-136)

Also the references need to be well standardised and seem not to follow any certain citing style/pattern especially those from online (internet) journals.

Authors Response: The references have been standardised and cited proper in the manuscript. (Page 23-29)

Reviewer #2

Study reports a very important aspect globally; and mostly in Africa where infertility carries important medical and mostly psychosocial implications. The analysis of outcomes has used standard methods, for both malaria and diabetes; results are very relevant and matched with the research questions. The design is appropriate, and the different essays have been conducted following the standardized methods. 

1. However, the application of the design presents some weaknesses which introduced doubt in the findings concerning introduction of possible bias.

Authors Response: Case-control studies just like other study designs have some limitations, a chief of these is recall bias. However, the inferential statistics was conducted using measured parameters. The possible limitations that may have presented weaknesses in the findings have been addressed in the limitation of the study. (Page 22, lines 395-401)

2. We suggest reworking on following observations and revised all criteria of a good case-control study to increase even more the quality of this study. The design of the study is quite good, and the conduct of the study was smoothly conducted. The technical aspects related to equipment’s lab techniques used by the author’s team was appreciable; however, the case-control aspect has been a bit neglected. Several aspects reported in case-control studies were not mentioned here (see methodology and even an aspect of the results) Using the checklist of reporting good case-control studies, we can realize for example:

• Assignment of patients in groups of the study?

Authors Response: Assignment of study participants in groups has been addressed in the study participants section. “The study participants comprised 254 consented adult males, 160 of them had registered and were receiving treatment at the diabetic clinic of the Effiduase Government Hospital and 94 control participants comprised of healthy blood donors, members of staff of EDH, and those visiting their relatives on hospital admission. Out of the 160 Type-2 diabetic men, 80 of them had only T2DM and the other 80 had T2DM co-morbid with malaria infection”. (Page 4, lines 81-85)

• Analysis not detailed, stratification-logistic regression, which assumptions?

Authors Response: The aim of this study sort to compare of means of the various sperm parameters within the three groups (T2DM only, T2DM & Malaria co-morbidity and the Control or the Healthy groups), hence the application of a One-way ANOVA and subsequently Tukey’s HSD post hoc tests for multiple comparisons of means of measured indices and not stratification-logistic regression.

• Socio-demographic data is not presented stratified by groups (case/controls) to get an idea about similarities/differences between the two groups

Authors Response: We have now presented the socio-demographic data in a stratified manner by groups namely; T2DM only, T2DM & Malaria co-morbidity and the Control or the Healthy groups).

Editor’s comment

Authors Response: Figure files are now in PACE format.

Editor’s comment: Please clarify the Tables' numbers uploaded in your PDF file.

Authors Response: The tables are numbered from “Table 1-4” and each table is found right underneath their description.

---

## [Editor Report · Decision Letter 1]

9 May 2023

Consequences of type-2 diabetes mellitus and malaria co-morbidity on sperm parameters in men; a case-control study in a district hospital in the Ashanti Region of Ghana.

PONE-D-23-06530R1

Dear Dr. Abdulai

We’re pleased to inform you that your manuscript has been judged scientifically suitable for publication and will be formally accepted for publication once it meets all outstanding technical requirements.

Kind regards,

Raquel Inocencio da Luz, Phd

Academic Editor

PLOS ONE

---

## [Editor Report · Acceptance letter]

18 Sep 2023

PONE-D-23-06530R1 

Consequences of type-2 diabetes mellitus and malaria co-morbidity on sperm parameters in men; a case-control study in a district hospital in the Ashanti Region of Ghana. 

Dear Dr. Abdulai:

I'm pleased to inform you that your manuscript has been deemed suitable for publication in PLOS ONE. Congratulations! Your manuscript is now with our production department. 

Kind regards, 

on behalf of

Dr Raquel Inocencio da Luz 

Academic Editor

PLOS ONE